# Influence of Lime on Strength of Structural Unreinforced Masonry: Toward Improved Sustainability in Masonry Mortars

Meera Ramesh [1], Manuel Parente [2], Miguel Azenha [2] and Paulo B. Lourenço [2,*]

1   Ryan Biggs Clark Davis, New York, NY 12065, USA
2   Department of Civil Engineering, University of Minho, ISISE, ARISE, 4800-058 Guimarães, Portugal
*   Correspondence: pbl@civil.uminho.pt

**Abstract:** The choice of a sustainable construction material needs to take into account not just the environmental impact of the material, but according to the 2030 Agenda for Sustainable Development by the UN, one also needs to consider ease of access, the utilization of locally available materials, and the durability and reliability of the construction itself. Mortared masonry has been used around the world for several hundred years as an accessible type of construction. In masonry mortars, lime and cement are often integrated together for combined advantages: enhanced workability, breathability, and better environmental performance due to the former, and higher strength and shorter setting duration due to the latter. However, despite being extensively studied for their effects on the mechanical properties of mortar, not much is known about the impact of varying lime and cement ratios in the binder on the mechanical performance of masonry as a whole. Variations in the properties of mortars do not always have a significant impact on the mechanical behavior of masonry structures. Therefore, this article details an experimental campaign to measure the compressive strength, E-modulus, flexural strength, and shear bond strength of masonry samples containing two distinct lime–cement mortars (1:2:9 and 1:1:6 cement:lime:sand) and one cement mortar (1:0:5). The results show that more than the presence of lime in the mortar, the strength of the mortar influenced the flexural strength of the masonry ranging from 0.1 to 1.2 MPa. No discernable correlation was observed between the presence of lime in the mortar and the cohesion in the masonry (0.29 to 0.41 MPa). The values of the compressive strength (6.0 to 7.2 MPa) and E-modulus (3.8 to 4.5 GPa) of the masonry decreased and pre-peak ductility increased with an increase in the quantity of lime in the mortar. The recommendations of Eurocode 6 for the flexural strength of the initial shear bond strength were found to be conservative for different mortar strength classes, and significantly unconservative for compressive strength (by 50% to 70%).

**Keywords:** compressive strength; flexural strength; shear bond strength; E-modulus; brick masonry; lime–cement mortars

## 1. Introduction

A holistic assessment of the sustainable development of infrastructure necessitates the consideration of different aspects, ranging from environmental to economic factors. In fact, Goal 9 of the 2030 Agenda for Sustainable Development of the United Nations [1], emphasizes the need for 'quality, reliable, sustainable' infrastructure. Additionally, it discusses 'affordable and equitable access' to such infrastructure via the enhancement of related scientific research. It is fairly well established that human beings have been using lime-mortared brick masonry for centuries, across the world, dating back to ancient Rome and Greece [2]. This makes masonry constructions a very commonly used, and, therefore, accessible, form of infrastructure. While different binders may be used in mortars, two of the most commonly used binding materials are lime and cement, which are often mixed together to obtain optimal performance [3].

From the viewpoint of environmental impact, lime-based construction materials as well as a few other binders have been fairly well researched, typically via life cycle assessments (LCA) [4–8]. LCA assesses sustainability through $CO_2$ emissions and the footprint of materials from their extraction to manufacturing, use in construction, and disposal. This also includes aspects related to the higher reutilization and recycling rates of lime-based construction materials, which can lead to decreased environmental pollution due to the otherwise necessary disposal of wastes. Yet, another important aspect affecting sustainability is serviceability behavior, durability, and reliability. Balanced decisions regarding the sustainability of construction materials can only be taken if their impact on mechanical performance at a structural level is well understood, which ultimately contributes to better serviceability behavior. In turn, improved serviceability behavior results in longer structural life cycles, and thus lower demand for natural materials and resources, thus contributing toward sustainability.

The focus of this research article, therefore, is on the mechanical performance of unreinforced brick masonry as a function of the type of mortar used in it, especially focusing on the presence of lime. The reason for this is that while lime/cement-mortared masonry is easily accessible and used across different geographical regions of the world, and there are existing studies that have focused on the environmental assessment of such materials [4–8], there is still limited research that studies the effects of lime at the masonry level. As illustrated in the following text, most research has focused on the impacts at the mortar level.

One of the most significant aspects of mechanical performance is compressive strength. Masonry compressive strength depends on aspects such as the type of adopted unit, the stiffness of the mortar, and the unit–mortar bond [9–20]. Despite the fact that the strength of masonry is a result of the combination of the strength of the unit and of the mortar, the unit typically has a greater impact on the strength of the masonry. A study by Lima et al. [21] indicates that in 10 mm and 15 mm thick mortar joints, a 150% increase in the strength of the mortar resulted in a 16% and 36% increase in masonry strength, respectively. Research conducted on concrete masonry prisms (both filled and unfilled ones), showed an increase of 35% in compressive strength stemming from a 250% increase in mortar strength [22]. In another study, increasing the compressive strength of mortar in unfilled prisms by approximately 72% resulted in an increase of no more than 20% in the strength of the masonry [23]. This means that the relationship between the strength of mortar and the strength of masonry is not linearly proportional, and the use of increasingly stronger mortars may not be effective in increasing the strength of masonry [24,25].

Notwithstanding the fact that mortar may not be the major contributor to the strength of masonry, it is an essential factor in determining its environmental impact [26–28], deformation properties, as well as non-linear behavior [29–32]. Environmentally, lime tends to outperform cement in several areas, some of the most noteworthy being human health and ecosystem quality. In addition, the recycling potential of lime is, to some extent, superior to that of cement, as the former is biodegradable and fully recyclable [23]. From the point of view of mechanical performance, especially focusing on deformation properties and non-linear behavior, it is also well established that an increase in the quantity of lime in the binder of a lime–cement mortar contributes to a lower value of mortar strength. With a reduction in mortar strength, the stress–strain behavior of masonry becomes significantly more non-linear [24]. Yet, considerable experimental data on its E-modulus assume that masonry behaves linearly up to 33–50% of its maximum compressive strength [22,24]. Limited studies have been conducted to evaluate the E-modulus of masonry through cyclic compressive loading. Since data from cyclic loading tests are not accessible for determining its E-modulus, the correlation of compressive strength with the secant E-modulus of masonry is most commonly used to gauge its stiffness [25]. Various global standards permit this, including the International Building Code (IBC) and Eurocode 6, both of which recommend the evaluation of Young's modulus to be at least 700 to 1000 times higher than the compressive strength, respectively [33,34]. Other sources have reported considerable



differences in these values, which can be anywhere between 80 and 1700 times the compressive strength [25,35,36]. The structural weakness of unreinforced masonry in the face of out-of-plane loads is a well-established issue, particularly during earthquakes [37–39]. Masonry's low tensile strength has been known to be the main contributor to this vulnerability [40–42]. Masonry is more resilient when subjected to bending forces perpendicular to the bed joints, as opposed to parallel to them. In the former case, the bond strength of head joints and the friction between them determine the resistance to compressive forces, while only the bond strength of bed joints contributes to resistance in the latter instance, as reported in [42]. Eurocode 6 [33] advises the use of the strength of mortar and the type of unit as indications to determine the values of characteristic flexural strength. Alternatively, experimental values stemming from four-point bending tests can also be employed [43]. Researchers have been investigating how various types, combinations, and reinforcements of mortar and masonry units can impact flexural strength [44–49]. This includes cement composites and FRP laminates. Nevertheless, the authors are not aware of any studies looking into the effect that the lime content in mortar has on the flexural strength and other characteristics of masonry [29,43].

In addition, the shear bond strength between masonry materials is a crucial parameter and has been examined by various researchers [10,50]. To evaluate its value, an experiment was conducted utilizing couplet or triplet specimens, with the latter being more prevalent among them [51–53]. Research has also been conducted to analyze the effect that pre-compression has on peak shear stress for brick masonry (wire-cut clay) and various cement mortars ranging from 10 to 30 MPa [54]. Experiments were performed to measure the shear strength of a variety of lime–cement mortar and concrete blocks [55]. The primary aim was to compare the relative strength of the mortar and unit. It was demonstrated that the strength of the mortar and the unit both contributed to improved cohesion as the amount of cement in the binder was increased. A different study corroborated the findings by analyzing lime–cement mortars with varying compositions (1:2:9, 1:1:6, and 4:1:12 cement:lime:sand) on different units, including molded clay brick, extruded clay brick, and concrete blocks in triplet specimens. The study concluded that the higher the cement content in the binder, the higher the shear strength across all kinds of units [56]. However, no research has been conducted on the coefficient of friction with pre-compression applied, denoting a gap in current knowledge. The shear strength and coefficient of friction values recorded for lime–cement mortars were in the ranges of 0.07 to 1 MPa and 0.80 to 0.92 MPa, respectively [55,56]. Additionally, the initial shear strength of lime-based mortars varied from 0.15 to 0.43 MPa [57].

This article examines the outcomes of experiments conducted on masonry specimens built with solid (frogged) molded clay bricks and three different mortar mixes (with 1:0:5, 1:1:6, and 1:2:9 as the volume ratios of cement:lime:sand). Specific attention was paid in this experimental campaign to pick mortar mix proportions that are already commonly used in the field by consulting with industry experts. A professional mason was hired to construct the masonry specimens as further illustrated in this article to ensure that the masonry was constructed in alignment with field practices, using workable mortars so that the conclusions of this research would be of practical value. An extensive and thorough experimental campaign was conducted to ensure repeatability. The properties of the masonry studied include compressive strength, flexural strength (parallel and normal to bed joints), the E-modulus via cyclic compression, and shear bond strength, including the parameters of cohesion and internal friction. All of these aspects when accounted for in totality directly align with Goal 11 of the 2030 Agenda for Sustainable Development of the United Nations, which focuses on 'strengthening efforts to safeguard the world's heritage', as well as 'building sustainable and resilient buildings utilizing local materials'.

## 2. Materials, Protocols, and Experimental Setups

### 2.1. Description of Raw Materials

Air lime and Portland cement were used as binders in the mixes. The cement used for this project was CEM I—42.5 R. As per EN 197-1 [58], CEM II, is allowed to have a higher variation in constituents in addition to clinker compared with its counterpart (CEM I). Some examples of constituents that fit this category include limestone filler, fly ash, silica fumes, calcined Pozzolana, and burnt shale. These constituents display high variability concerning their compositions, which is a function of aspects such as the treatment method or source of origin. Consequently, CEM I was selected in order to reduce the variability mentioned above. The technical data sheet of the batch of cement CEM I—42.5 R used for this study was supplied by Secil (ACM-040/2016). The material had a density of 3.12 g/cm$^3$ and a Blaine specific surface of 3508 cm$^2$/g, while the clinker composition was 12.6% C2S and 62.2% C3S. Table 1 outlines the chemical compositions of the primary components, with LOI denoting ´loss on ignition´. The LOI value was obtained from manufacturer according to the recommendations prescribed in EN 459-2 [59]. The measured apparent bulk density stood at 0.93 g/cm$^3$. Concerning the type of air lime, CL90-S was adopted in this experimental campaign. Similar to the choice of cement, variability was one of the main concerns driving the selection of the lime. According to EN 459-1 [60], CL 90-S is the air lime with the least variation in chemical composition and the highest percentage of available lime (at least 80% by mass). The material originated from Lhoist, and its specifications were obtained from its datasheet (control number 90000998782). The BET surface area of the particles was recorded as 150,000 cm$^2$/g, while their density was calculated as 2.24 g/cm$^3$. Furthermore, the distribution of particle sizes averaged between 5.5 and 6.5 μm. Table 1 also describes the chemical composition of the lime based on results from X-ray fluorescence (Axios Panalytical), expressed as oxide equivalent. LOI was also measured by the manufacturer according to EN 459-2 [59]. The apparent bulk density of the lime was 0.36 g/cm$^3$.

**Table 1.** Chemical compositions of binders used (CEM I—42.5 R and CL 90—S).

| Material | LOI (%) | MGO (%) | SO$_3$(%) | AL$_2$O$_3$ (%) | FE$_2$O$_3$ (%) | K$_2$O (%) | SIO$_2$ (%) | CAO (%) |
|---|---|---|---|---|---|---|---|---|
| CEMENT CEM I-42.5 R | 2.05 | 1.75 | 3.05 | 4.27 | 3.2 | 0.77 | 20.55 | 63.4 |
| LIME CL 90-S | 25 | 0.68 | 0.197 | 0.06 | 0.05 | 0.013 | 0.12 | 74.35 |

Different options were considered regarding the selection of the unit for masonry construction. Apart from references from the literature [61–66], discussions with members of the industry [58] were also accounted for. From these, it was inferred that though different options were available in the market [66–68] (including solid or hollow concrete blocks, calcium silicate blocks, and autoclaved aerated concrete blocks), clay bricks would be the ideal unit to begin the testing of lime–cement mortars. Clay bricks can be divided into sub-groups depending on their geometry, perforations (horizontal or vertical), and the method of manufacture (extruded or molded) [69]. Ultimately, the options were narrowed down to solid molded clay bricks. The main criteria driving this selection were the water absorption and the initial rate of absorption (IRA) of this type of brick. The expectation was that bricks with higher IRAs and absorption levels (generally molded instead of wire-cut/extruded bricks) could help determine the variance in mortar binders via the mechanical behavior of the masonry.

Consequently, a solid-clay frogged molded brick of dimensions 215 × 102 × 65 mm$^3$ provided by Wienerberger was chosen for this project. According to the DoP number 152110-B1W1210 from the manufacturer, this brick type falls under category 1, tolerance T1, and range R1. Furthermore, according to EN 771-1 [69], its frogged volume is less than 20%.

### 2.2. Mechanical Characterization of Unit

Table 2 shows the results from a lab test of the selected brick in terms of the compressive strength, E-modulus, IRA, and water absorption. The values of compressive strength were supplemented with the corresponding coefficient of variation. The average value of the brick's compressive strength was 26.2 MPa. Following the guidelines of EN 772-1 [70], a shape factor was used to calculate the normalized compressive strength from its average value. This factor differed depending on each specimen's height-to-width ratio (ranging from 0.55 to 0.66), which, in turn, was determined using a caliper. As per EN 771-1 [69], all the dimensions of the bricks fell within tolerance category T1. Table 2 displays the mean value of the normalized compressive strength.

**Table 2.** Mechanical characterization of bricks used to construct masonry.

| Property | Compressive Strength * (MPa) | | E-Modulus (GPa) | Flexural Strength (MPa) | Water Absorption (%) | IRA (kg/m²·min) |
|---|---|---|---|---|---|---|
| | Complete Brick | Cubic Specimen | | | | |
| Average (CoV %) | 22.03 (22.7%) | 21.15 (13.7%) | 4.9 (15.7%) | 5.41 (21.0%) | 10.3 (7.6%) | 3.55 (15.6%) |

* Normalized value obtained by multiplying average value by shape factor according to EN 772-1 [70].

### 2.3. Mortar Mixes: Preparation and Characterization

As previously mentioned, three mortar mixtures were adopted for this experimental campaign, namely, 1:0:5, 1:1:6, and 1:2:9 (cement:lime:aggregate proportions expressed by volume), as shown in Table 3. The label 'Ref' is used to indicate the reference mortar mix, while 'L50' and 'L67' stand for the mortars with 50% and 67% of lime as the binder, respectively. Note that the letter 'L' refers to the amount of lime in each variation of mortar. The water–binder ratio was carefully chosen such that all mortar mixes had a flow table value of 175 ± 10 mm, which would facilitate the use of the mortar by masons. Great care was taken to ensure that the mix compositions chosen in this campaign were representative of what is commonly used in the field and were also based on consultations with members of the industry [58]. The rationale behind the reference mix was to have a widely used 'standard cement mix' with no lime in it to understand the impact of the absence of lime (Table 4). Furthermore, L50 and Ref were found to have similar compressive strengths at the mortar level, so the main difference between the two was the presence of lime. Between L50 and L67, the main point of comparison was the strength of the mortar since both of them had a significant quantity of lime present in them (Table 4).

**Table 3.** Compositions of mortars used in masonry.

| Nomenclature | Cement:Lime:Sand (Volume) | Cement (kg) | Lime (kg) | Aggregate (kg) | | w/b Ratio (by Weight) | w/b Ratio (by Volume) |
|---|---|---|---|---|---|---|---|
| | | | | Sand | Filler | | |
| Ref | 1:0:5 | 233.5 | 0 | 1743.6 | 206.4 | 1.2 | 1.12 |
| L50 | 1:1:6 | 192.6 | 73.4 | 1726.1 | 204.4 | 1.09 | 0.70 |
| L67 | 1:2:9 | 128.4 | 97.9 | 1726.1 | 204.4 | 1.3 | 0.71 |

**Table 4.** Comparison of mechanical strengths of mortars used in masonry specimens in standard and in situ conditions at 28 and 90 days of curing.

| Property Condition | Compressive Strength (MPa) (CoV %) | | | | Flexural Strength (MPa) (CoV %) | | | | E-Modulus (GPa) (CoV %) | |
|---|---|---|---|---|---|---|---|---|---|---|
| | In Situ | | Standard | | In Situ | | Standard | | In Situ | Standard |
| Age | fc-28 | fc-90 | fc-28 | fc-90 | ff-28 | ff-90 | ff-28 | ff-90 | E-Mod-90 | E-Mod-90 |
| L67 | 4.12 (3.4) | 5.30 (5.2) | 4.35 (10.8) | 4.69 (2.1) | 1.57 (3.5) | 1.95 (2.1) | 1.88 (5.0) | 1.88 (4.5) | 7.90 (5.3) | 8.54 (1.2) |
| L50 | 9.75 (7.6) | 10.07 (8.5) | 9.35 (5.1) | 9.28 (5.7) | 2.99 (9.3) | 3.55 (7.8) | 3.93 (0.7) | 3.42 (3.5) | 15.97 (17.5) | 14.86 (2.2) |
| Ref | 10.88 (8.9) | 12.08 (6.0) | 10.27 (7.3) | 11.21 (2.7) | 3.04 (1.0) | 3.78 (11.3) | 3.14 (2.4) | 3.53 (7.1) | 15.21 (5.1) | 19.47 (10.5) |

Based on recommendations found in the literature, the adopted thickness for the mortar joints was 10 mm [24,25,71–74]. The masonry specimens were built and air-cured in the basement of the laboratory at the University of Minho. To make sure the masonry specimens were moved safely, a manual lift and an elevator were used to transport them from the basement to the ground floor of the laboratory for testing (which was one level above). The process was conducted with great care, involving two people to ensure no detriment was caused to the specimens. This guaranteed that damage and microcracking could be avoided. Temperature and humidity levels in both the testing and storage areas were monitored for one year and proved to be in close proximity, approximately $21 \pm 1\,^\circ C$ and $70 \pm 5\%$ RH, respectively. Before the preparation of the masonry specimens, the binders, bricks, sand, filler, and water were all stored at the same location.

In accordance with RILEM LUM B1 [75], all batches of mortar were used within 60 min of preparation. The adopted starting time reference (t = 0) corresponded to the moment when the binders came into contact with water. Approximately 13 kilos of aggregates was used for each prepared batch of mortar, ensuring that the mortar casting was carried out in less than one hour. A professional mason working with the laboratory was responsible for putting together the masonry samples. Measurements of all materials, including water, binder, and aggregates, were accomplished by taking weight as a reference to increase the accuracy in the mix designs. At the start of the process (t = 0), the mixture was placed in a 50 cm × 50 cm barrel and mixed for approximately 90 s, with the aggregate being added between the 30 to 90 s timeframe. After mixing for 90 s, the process was halted to allow the scraping of mortar from the sides and bottom of the barrel. Thereafter, further mixing continued for a period of 90 s. The type of mixers used in the field by masons are comparable to the Parkside PFMR 1400 B1 used at the laboratory, which is equipped with a powerful 1400 W motor and 700 rpm rotation speed [76]. It also has a stirrer holder M14, making it suitable for mixing large volumes. At the same time, clay bricks were dusted off and placed underwater for half an hour before use in order to prevent them from absorbing moisture from the mortar mix, which would otherwise affect the masonry bond strength [75]. To enable a comparison between their performances, two different curing conditions were adopted for the mortars:

(a) In situ mortars: The mortars Ref, L50, and L67 were cast in large batches as described above. From these batches, small quantities of mortar were separated during preparation for masonry construction and cured next to the masonry specimens, thus subjecting them to the same temperature and humidity conditions. This set was designated as 'in situ mortars'.

(b) Standard mortars: The three mixes (Ref, L50, and L67) were made in accordance with the procedures suggested by European standards [77,78]. For the initial 7 days, all lime–cement mortar samples were kept in conditions of $20 \pm 1\,^\circ C$ and $95 \pm 5\%$ relative humidity. Afterward, the conditions were adjusted to a temperature of $20 \pm 1\,^\circ C$ and $65 \pm 5\%$ relative humidity until the specimens were tested. Per the standards outlined in EN 1015-11 [78], specimens of the L50 and L67 lime–cement mortars were de-molded after two days since the lime content in their binder was lower than 50% by mass. For the Ref mix specimens, there was a curing process in accordance with EN 196-1 [77]. This process entailed placing the molds containing the Ref mix in a plastic bag for 24 h, at a temperature of $20 \pm 1\,^\circ C$ and relative humidity (RH) of $95 \pm 5\%$, inside a climatic chamber. Thereafter, the Ref samples were submerged in water at a temperature of $20 \pm 1\,^\circ C$, up to the time of testing.

EN 1015-11 [78] was used as a reference to ascertain the flexural and compressive strength of mortars cured under two different conditions. This research team has previously published a different article, which focuses solely on the mechanical characterization of 15 mortars mixes, including those featured in this paper [79]. Table 4 showcases the outcomes of the mechanical characterization of the 3 mixes discussed in this paper. In this table, it is possible to observe an interesting coherence between the mechanical properties ($f_c$, $f_{ct}$,

and fl) of the mortars cast using the two different mixing protocols (hence consubstantiating their mutual correspondence).

## 3. Experimental Setup

The compressive strength of the masonry was evaluated in accordance with the guidelines specified in EN 1052-1 [80]. Specimens for the experiment consisted of a single leaf of bricks, with a height of 450 mm (6 rows of bricks) and a width of 440 mm (2 bricks per row), as illustrated in Figure 1. In order to evaluate the effectiveness of the specimens, they were allowed to cure for 90 days prior to testing. A hydraulic actuator with a load capacity of 1000 kN was utilized, resorting to the displacement control technique at a speed of 3 μm/s. To assess deformation in the specimens, four LVDTs were used in total. These were situated on the wider faces of each sample (2 at the front and 2 at the back), as illustrated in Figure 1. The final compressive strength was determined by testing 3 masonry specimens for each type of mortar. Before testing, one masonry specimen for each mortar type was tested to confirm conditions such as the capacity of the actuator, the testing speed, and the value of cyclic loads to be applied to the specimens for E-modulus measurements. These specimens were not taken into account for the calculation of results.

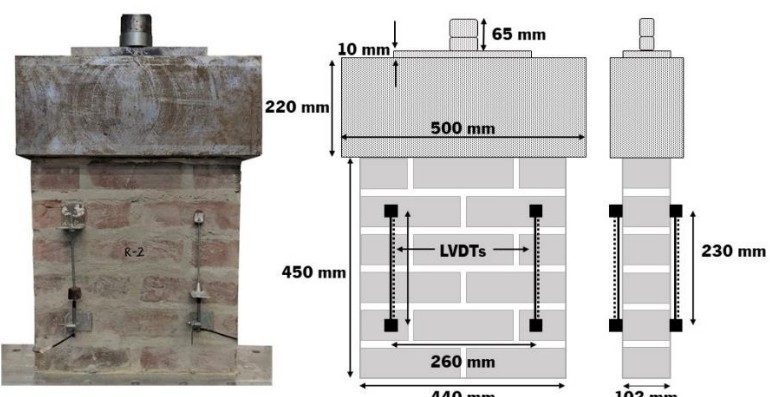

**Figure 1.** Schematic representation of set-up used for testing compressive strength and E-modulus of masonry specimens.

With regard to flexural strength, the masonry specimens were tested in both directions ('parallel', referring to parallel to the bed joints, and 'perpendicular', referring to perpendicular to the bed joints) according to the recommendations of EN 1052-2 [43] (Figure 2). To conduct the tests, a 300 kN hydraulic actuator was used with displacement control at a rate of 3 μm/s. In both parallel and perpendicular cases, 2 LVDTs set up on either side of each specimen were used to measure the out-of-plane deformation. In the case of the parallel layout, the single-leaf specimens had 9 courses of bricks, with 2 bricks in each course. The size of each sample was measured as 670 mm in height and 440 mm in width (refer to Figure 2). EN 1052-2 [43] outlines the recommended dimensions of the length of a specimen in the direction of the span ($l_s$), the spacing between supports ($l_1$), and the distance between inner bearings ($l_2$) for the proper application of a load. The values adopted for this case were $l_s$ = 670 mm, $l_1$ = 570 mm, and $l_2$ = 290 mm. For the perpendicular case, while the specimens were also single-leaf specimens, they featured 6 courses of bricks, with 4 bricks in each course. The dimensions of each sample were 890 mm in height and 440 mm in width (Figure 2). In this case, $l_s$ corresponded to 890 mm, while $l_1$ and $l_2$ were 790 mm and 380 mm, respectively.

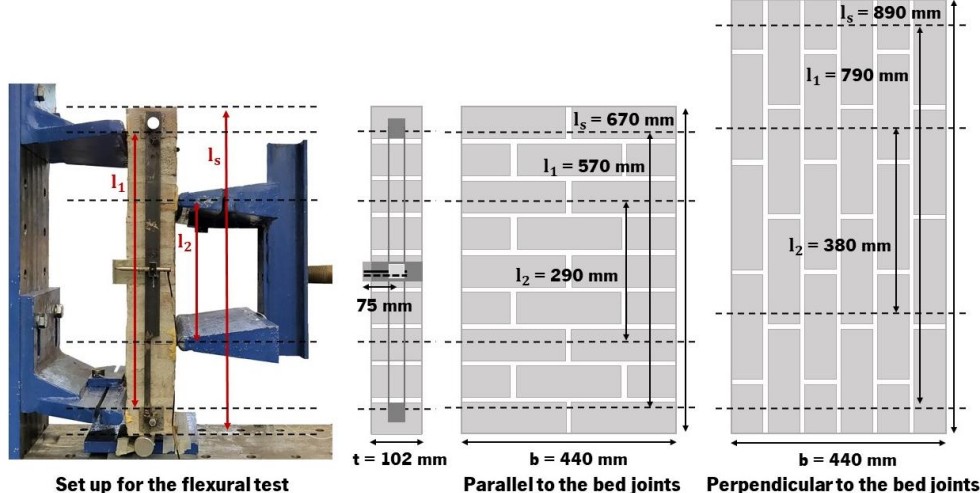

**Figure 2.** Schematic representation of set-up used for testing flexural strength of masonry specimens, in directions parallel and perpendicular to bed joints.

The shear bond strength of the samples was assessed per the guidelines of EN 1052-3 [81]. Nine masonry specimens (triplets) were tested for each type of mortar, resulting in a total of twenty-seven specimens evaluated (Figure 3). For each mortar, three specimens were tested for three levels of perpendicular pre-compression: 0.2 MPa, 0.6 MPa, and 1 MPa. A manually operated hydraulic pump was used, with a capacity of 100 kN to maintain a steady pre-compressive load. This load had an average maximum variation of ≤0.9 kN across the different tests. To apply the shearing load, an additional actuator with a 200 kN force and a sampling rate of 4 Hz was employed. This actuator was capable of providing the desired displacement control at 3 μm/s. Two LVDTs were positioned on either side of the specimen (front and back) so as to measure the amount of movement between the bricks and mortar joints (as seen in Figure 3). This enabled the accurate measurement of the relative slip between bricks.

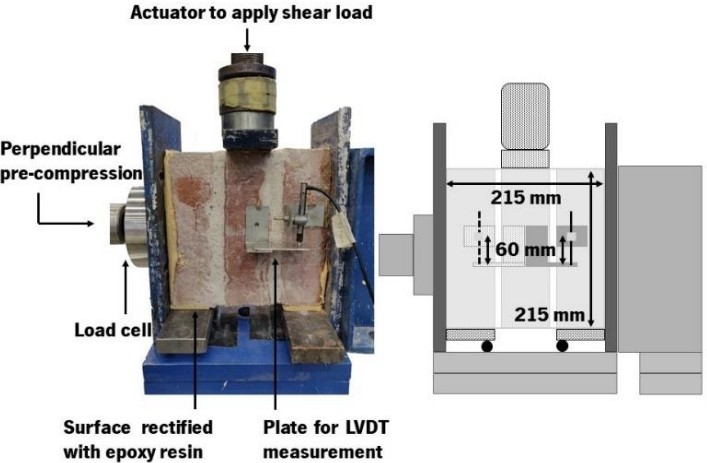

**Figure 3.** Schematic representation of set-up used for testing shear bond strength of specimens.

## 4. Results and Discussion

### 4.1. Compressive Strength and E-Modulus

The results of testing the masonry wallets with different types of mortars with regard to compressive strengths and E-moduli can be seen in Table 5 and Figure 4, along with the corresponding error bars. Amongst all the masonry mortars, Ref has the highest compressive strength. This is followed by L50 and then L67. The masonry strength was seen to increase with the strength of the mortars (Ref, L50, and L67). According to EN

1052-1 [80], the characteristic values of compressive strength were estimated by taking the mean compressive strength of masonry and dividing it by a factor of 1.2. The E-modulus-to-compressive-strength ratio was between 600 and 650 for the average value of strength and 725 to 775 for the characteristic value. There was no significant trend to observe in this regard.

**Table 5.** Compressive strengths and E-moduli of masonry wallets.

| Mortar Type | Compressive Strength (MPa) | | Coefficient of Variation (%) | E-Modulus (GPa) | Coefficient of Variation (%) | E-Mod/fc | |
|---|---|---|---|---|---|---|---|
| | Average | Characteristic | | | | Average | Characteristic |
| L67 (1:2:9) | 6.02 | 5.01 | 6.9 | 3.88 | 11.3 | 645 | 774 |
| L50 (1:1:6) | 6.65 | 5.54 | 11.8 | 4.02 | 12.5 | 605 | 726 |
| Ref (1:0:5) | 7.18 | 5.98 | 6.1 | 4.46 | 19.3 | 621 | 746 |

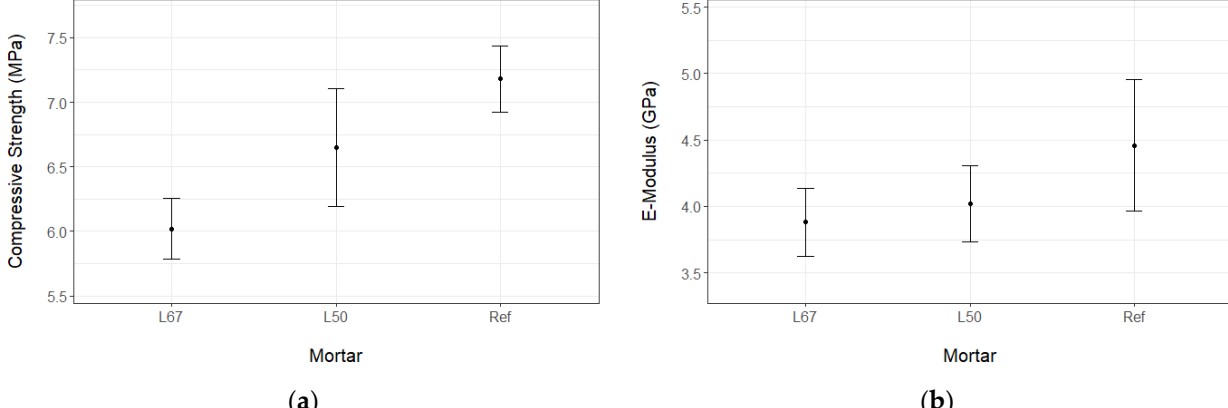

(**a**)　　　　　　　　　　　　　　　(**b**)

**Figure 4.** (**a**) Compressive strength and (**b**) E-moduli of masonry wallets, including error bars.

As all the masonry specimens were made using the same type of brick, the varying behavior must be attributed to the type of mortar used. To better understand the influence of the mortar on masonry strength, the strength of the mortar measured under 90-day standard conditions (Table 4) was compared with the compressive strength of the masonry (Table 5):

(a)　Ref and L50: The difference in compressive strength was 17% at the mortar level, while the difference in strength at the masonry level was only 7%.

(b)　Ref and L67: At the mortar level, the difference in compressive strength was 58%, while at the masonry level, this difference was only 16%.

This indicates that while the strength of the masonry and mortar are interdependent, their relationship is not linear. In fact, as demonstrated in Figure 4, a substantial increase in the strength of the mortar yields only a marginal increase in the strength of the masonry. Eurocode 6 [33] corroborates this, stating that the compressive strength of a masonry structure is impacted by the compressive strength of its mortar by an exponent value of 0.3. Thus, according to this standard, the masonry strengths of L50 and L67 would decline by 5% and 23%, respectively, in comparison with Ref.

Similar to compressive strength, the E-moduli showed decreasing values from Ref to L50 and L67. Additionally, the stiffness of the mortar determined on day 90 (see Table 4) was also compared with that of the masonry specified in Table 5:

(a)　Ref and L50: The difference in the E-moduli was 24% at the mortar level, while the difference in the E-moduli at the masonry level was only 10%.

(b)　Ref and L67: At the mortar level, the difference in the E-moduli was 56%, while at the masonry level, it was only 13%.

Similar to what was observed for compressive strength, it appears that attaining an increase in the value of the E-modulus of masonry would require the use of a mortar with

significantly higher stiffness (Figure 5). Once again, this aligns with the recommendations of Eurocode 6 [33], which states that the E-modulus of masonry is linearly related to its compressive strength, with a multiplication factor of 1000 for the characteristic value of strength, or a factor of 830 for the corresponding average value, which is in the same ballpark as the values found herein, which are in the range of 725–775 (Table 5).

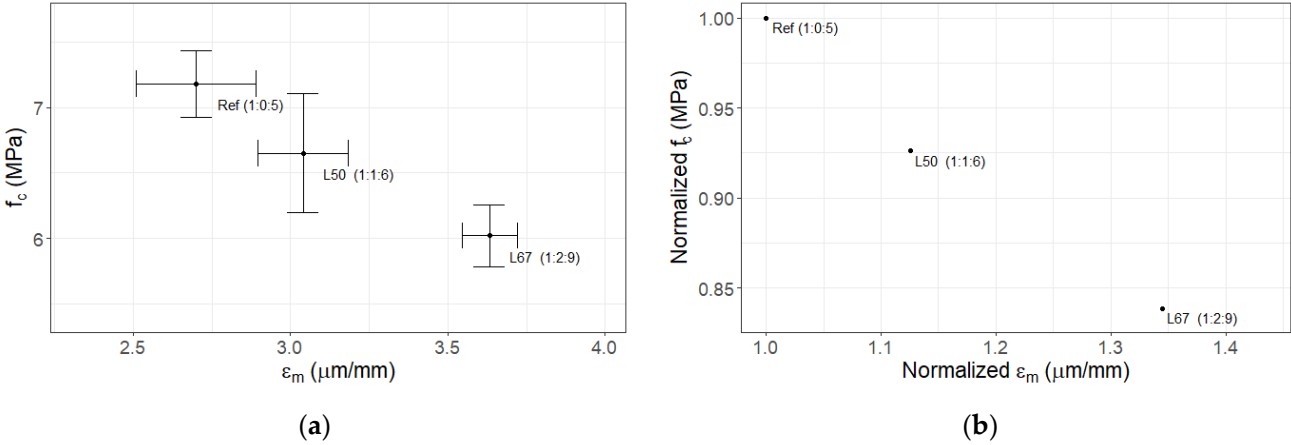

(**a**)          (**b**)

**Figure 5.** Vertical strain at peak stress vs. maximum compressive strength of masonry: (**a**) absolute values; (**b**) normalized values.

Aiming to compare the experimentally obtained values for the characteristic compressive strength of masonry, with the recommendations in Eurocode 6 [33], the values of the compressive strength of mortar and brick were fed into the equation $f_k = K f_b{}^\alpha f_m{}^\beta$ (Table 5). Here, $f_k$ corresponds to the characteristic compressive strength of masonry, $f_b$ to the compressive strength of brick, and $f_m$ to the compressive strength of mortar. The values of 0.7 and 0.3 were adopted for the $\alpha$- and $\beta$-coefficients, respectively, per the recommendations in Eurocode 6 [33]. According to the values predicted with this standard, which are represented in Table 6, the compressive strength of masonry was consistently overestimated in the 50% to 70% range. However, if the comparison was performed with the mean values of compressive strength (Table 5), as opposed to the aforementioned characteristic values of compressive strength, the difference between the Eurocode 6 predictions and the experimental values fell within a range of 27–41%. Kaushik et al. [25] conducted experimental studies to determine the performance of clay brick masonry in compression, with different strengths of mortar and brick being tested. Based on their findings, they proposed values for K, $\alpha$, and $\beta$ via regression analysis. This allowed them to effectively predict the average experimental strength of masonry from nine distinct research studies. The research revealed that the estimates were accurate up to an error rate of 40% when it came to evaluating a brick strength of less than 26 MPa. Since the strength of brick used in this research was also less than 26 MPa, the same values of K, $\alpha$ and $\beta$ are shown in the second set presented in Table 6. From the table, one can easily infer that those values consistently underestimate the strength of masonry obtained in this research in the range of 29% to 37%. Although the predictions are comparable to those of Eurocode 6, they were more conservative.

**Table 6.** Values of coefficients for $f_k$ and comparison with Eurocode 6.

| K | α | β | Mortar | Masonry Strength (MPa) | | Difference (%) |
|---|---|---|---|---|---|---|
| | | | | **Predicted** | **Experimental** | |
| 0.55 | 0.7 | 0.3 | Eurocode 6 [33] | Experimental (characteristic value) | | |
| | | | L67 | 7.65 | 5.01 | 52.5 |
| | | | L50 | 9.38 | 5.54 | 69.4 |
| | | | Ref | 9.93 | 5.98 | 66.0 |
| 0.55 | 0.7 | 0.3 | Eurocode 6 [33] | Experimental (average value) | | |
| | | | L67 | 7.65 | 6.02 | 27.0 |
| | | | L50 | 9.38 | 6.65 | 41.1 |
| | | | Ref | 9.93 | 7.18 | 38.3 |
| 0.63 | 0.42 | 0.32 | Kaushik et al. [25] | Experimental (average value) | | |
| | | | L67 | 3.79 | 6.02 | −37.0 |
| | | | L50 | 4.72 | 6.65 | −29.0 |
| | | | Ref | 5.02 | 7.18 | −30.2 |

As shown in Figure 5, the vertical strain associated with the highest compressive strength of masonry wallets was compared with the actual maximum compressive stress. The figure illustrates absolute (Figure 5a) and relative (Figure 5b) values that have been normalized with regard to the maximum deformation/vertical strain of the reference mix, as well as the associated error bars. When it comes to a comparison with Ref, L50 typically showcases 7.4% lower strength and 13.7% more strain. L67 has 16.2% less strength compared with Ref and 33.7% more strain. The strength of L67 is also 9.5% lower and its strain is 17.6% higher than that of L50. All these values depict that with every 1% reduction in the masonry wallet's strength, the strain at its highest point appears to increase by approximately 2%.

*4.2. Flexural Strength—Parallel and Perpendicular to the Bed Joints*

Table 7 demonstrates that the flexural strength of masonry in both directions (parallel and perpendicular to bed joints) is affected by the strength of the mortar. The L67 mix featuring the least strength at the mortar level also leads to the lowest flexural strength of masonry in both perpendicular and parallel directions. The two mixes L50 and Ref, both exhibiting similar strengths at the mortar level, seem to yield comparable values of masonry flexural strength. Studies have also shown an increase in masonry flexural strength with mortars of higher compressive strengths [82]. It was not possible to draw any conclusion with regard to the presence of lime in the binder, since L50 and Ref show slightly higher values in the parallel and perpendicular directions, respectively.

**Table 7.** Values of flexural strength of masonry in the directions parallel and perpendicular to the bed joints.

| Mortar Type | Parallel (MPa) | (CoV %) | Perpendicular (MPa) | (CoV %) |
|---|---|---|---|---|
| L67 | 0.10 | 18.4 | 0.78 | 5.7 |
| L50 | 0.23 | 3.8 | 1.11 | 3.8 |
| Ref | 0.19 | 8.5 | 1.12 | 14.0 |

According to Eurocode 6 [33], the recommended values of characteristic flexural strength of masonry are based on the type of unit used and the strength of the mortar. In this research, the units' category was clay bricks, and the mortar categories were ≥5 MPa for Ref and L50, and <5 MPa for L67 (Table 4). Additionally, the flexural strength data from this research (Table 7) are presented using average values. Yet, to make an accurate comparison with Eurocode 6 recommendations, characteristic values are required.

Accordingly, the experimental values were divided by a factor of 1.5 as per the guidelines in EN 1052-2 [43], as presented in Table 8. Note that the flexural strength of masonry prescribed by Eurocode 6 [33] is based on the compressive strength of mortars tested in standard conditions. However, should in situ tests for mortars at 90 days be used as a reference, all mortars would fall in the category of $\geq 5$ MPa (Table 4).

**Table 8.** Characteristic values of flexural strength and recommendations of Eurocode 6.

| Mortar Type | Compressive Strength of Mortar (MPa) | | Flexural Strength—Parallel (MPa) | | Flexural Strength—Perpendicular (MPa) | |
|---|---|---|---|---|---|---|
| | Experimental | In Eurocode 6 | Experimental | Eurocode 6 | Experimental | Eurocode 6 |
| L67 | 4.69 | <5 | 0.06 | 0.1 | 0.52 | 0.2 |
| L50 | 9.28 | $\geq 5$ | 0.15 | 0.1 | 0.74 | 0.4 |
| Ref | 11.21 | $\geq 5$ | 0.13 | 0.1 | 0.75 | 0.4 |

As per Eurocode 6 [33], the flexural strength in the direction of level bed joints should be considered 0.1 MPa, regardless of the mortar's strength. It is clear that the L67 mortar has a flexural strength of less than 0.1 MPa, while those of L50 and Ref are higher. In the perpendicular direction, all the mortars yield better flexural strength values than what is recommended by the code.

### 4.3. Shear Bond Strength and Cohesion

Table 9 shows that with increased perpendicular pre-compression, the maximum shear capacities of all specimens will also predictably increase. It appears that while the masonry with mortar L67 exhibits lower values of shear stress, specimens with mortars Ref and L50 display comparable shear stress levels under the same levels of pre-compression. Table 9 showcases the maximum shear stress with the coefficients of variations for each type of mortar when subjected to 0.2 MPa, 0.6 MPa, and 1 Mpa vertical/normal pre-compression, respectively. The maximum shear stress for the analysis ($\tau$) is an average of three distinct sample measurements for the corresponding pre-compression level.

**Table 9.** Values of maximum shear stress obtained for masonry specimens with varying levels of vertical pre-compression.

| Shear Stress—$\tau$ (MPa) Mortar | Vertical Pre-Compression/Normal Stress—$\sigma$ (MPa) | | | | | |
|---|---|---|---|---|---|---|
| | 0.2 | CoV % | 0.6 | CoV % | 1.0 | CoV % |
| L67 | 0.37 | 9.8 | 0.73 | 15.9 | 0.83 | 4.0 |
| L50 | 0.43 | 14.7 | 0.87 | 7.3 | 1.12 | 8.9 |
| Ref | 0.54 | 8.2 | 0.89 | 7.2 | 1.11 | 7.5 |

To determine the values of parameters like the coefficient of friction and cohesion, the normal stress (or vertical pre-compression) was plotted against the shear stress for different mortar types (as illustrated in Figure 6). Moreover, to make sure the Mohr–Coulomb criterion could be implemented, it was necessary to validate the linear relationship between the normal force and shear stress for each type of mortar. Linear regressions were carried out on the data and yielded $R^2$ values of 0.99, 0.97, and 0.90 for the Ref, L50, and L67 mortars, respectively, thus validating the use of the Mohr–Coulomb equation. Table 10 outlines the values of different parameters with respect to each type of mortar. It presents $\tan(\Phi)$ as the coefficient of friction, c is the cohesion or initial shear stress, $\tau$ represents the shear stress, and $\sigma$ denotes the normal stress or vertical pre-compression.

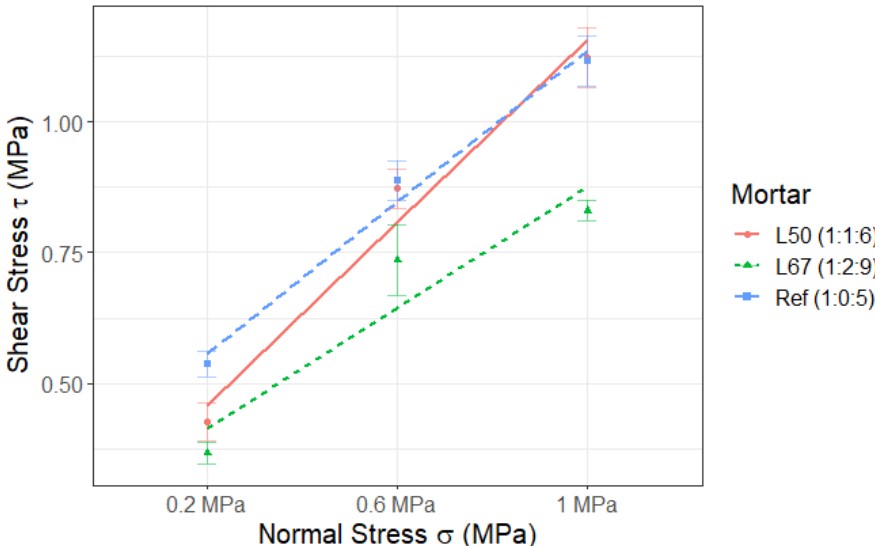

**Figure 6.** Normal stress versus shear stress for masonry triplets with different mortars, including error bars.

**Table 10.** Joint characteristics of different types of mortars used with brick masonry.

| Type of Mortar | Cohesion (MPa)—c | Characteristic Shear Strength (MPa)—$f_{vko}$ | Angle of Friction (Φ) | Coefficient of Friction (tan Φ) | Characteristic Coefficient of Friction (tan Φ) |
|---|---|---|---|---|---|
| L67 | 0.30 | 0.24 | 30.0 | 0.58 | 0.46 |
| L50 | 0.29 | 0.23 | 41.0 | 0.87 | 0.69 |
| Ref | 0.41 | 0.33 | 35.8 | 0.72 | 0.58 |

According to EN 1052-3 [81], the shear strength ($f_{vko}$) and coefficient of friction (tan(Φ)) were determined by taking the average of the results and multiplying it by a factor of 0.8. As per Eurocode 6 [33], the values of cohesion and initial shear strength ($f_{vko}$) can be determined using the compressive strength of the mortar class and the type of unit. According to this standard, mortars with strengths in the 2.5–9 MPa range (L67 and L50) should have a cohesion value of 0.2 MPa, while those with strengths between 10 and 20 MPa (Ref) should have a value of 0.3 MPa. Analyzing Table 10, it is evident that all the mortars have cohesion values that surpass the recommendation stated in Eurocode 6 [33]. The data collected during this research did not indicate any particular trend regarding cohesion (Table 10). Both L50 and L67 had lower cohesion values compared with Ref, while the coefficient of friction of L50 was higher than that of both.

Previous research indicates that characteristics of bricks such as surface roughness, absorption rate, and mechanical strength might also have a direct effect on the cohesion and internal friction values apart from the mortar type [51,54,83]. Thus, further experiments need to be conducted in order to fully understand these factors. Table 11 demonstrates that L50 shows a greater maximum shear stress capacity than the L67 mixture across all pre-compression levels. Concurrently, Ref displays higher readings compared with L50 at 0.2 and 0.6 MPa of pre-compression, with approximately the same results at 1 MPa. Considering that L50 has a higher strength at the mortar level in comparison with L67, whereas Ref exhibits a slightly better compressive strength than L50, this seems to suggest that the compressive strength of the mortar can influence the maximum shear bond stress in the masonry. A graphical representation of this is seen in Figure 7, which depicts the maximum shear stress of the masonry based on the compressive strength of the mortar used and its corresponding error bars. This relationship may be confirmed by the literature as well, in which a mortar with a higher compressive strength led to a higher shear bond strength in triplet specimens [16].

**Table 11.** Values of cohesion obtained for different types of mortars, using the same coefficient of friction for all.

| Type of Mortar | Cohesion (MPa)—c | Characteristic Shear Strength (MPa)—$f_{vko}$ | Angle of Friction (Φ) | Coefficient of Friction (tan Φ) |
|---|---|---|---|---|
| L67 | 0.21 | 0.17 | | |
| L50 | 0.30 | 0.24 | 40.4 | 0.85 |
| Ref | 0.34 | 0.27 | | |

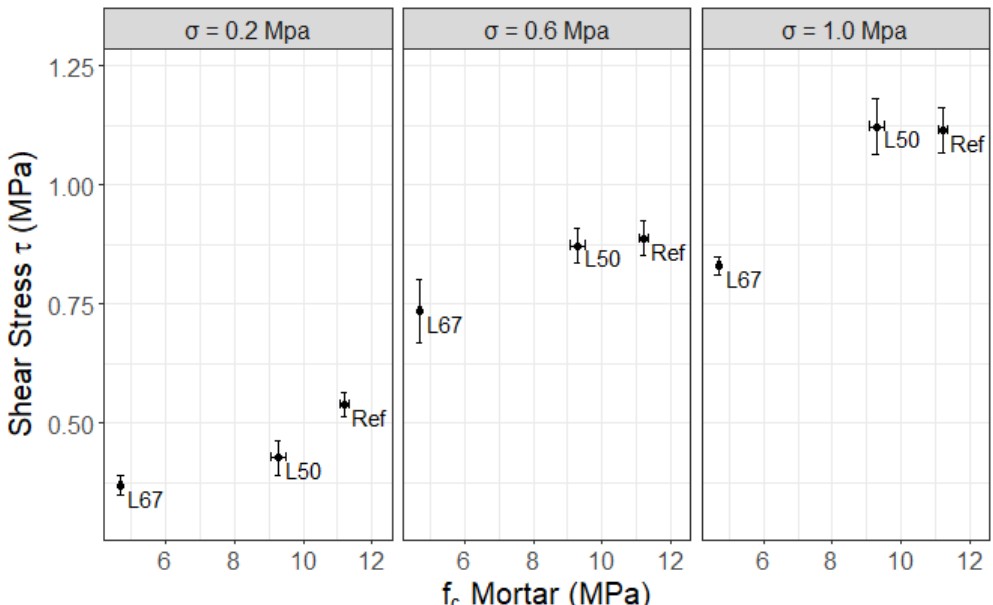

**Figure 7.** Maximum shear stress of masonry triplets as a function of compressive strength of mortar.

Table 11 indicates that both L67 and L50 have similar levels of cohesion, which are lower than the value of Ref. This is contrary to expectations, as the compressive strength of the mortar is frequently used as an indication of cohesion, for instance, in Eurocode 6 [33]. Bearing in mind that both mortars L50 and Ref have similar compressive strength (in standard conditions), ranging between 9.28 and 11.21 MPa, and that L67 has a lower compressive strength (4.69 MPa), one would expect the cohesion of L50 and Ref to be in the same range, while that of L67 to be lower. The linear regression analysis of the various mortars in Figure 6 shows that L67's maximum shear strain at 1 MPa is lower than anticipated, despite no irregularities observed in the specimens or testing conditions while conducting the experiments. A linear regression analysis was done (Figure 8) considering all the data from Table 9, except one datapoint of shear stress of L67 at 1 MPa, to interpret the data without the anomaly. The R-squared value obtained was 0.94 and the coefficient of friction was 0.85. These values allowed for the determination of the cohesion for each of the mortars (L67, L50, and Ref) which were found to be 0.21 MPa, 0.30 MPa, and 0.34 MPa respectively as seen in Table 11. This trend is consistent with the values of compressive strength of the mortars as well as the flexural strength of masonry (Table 7).

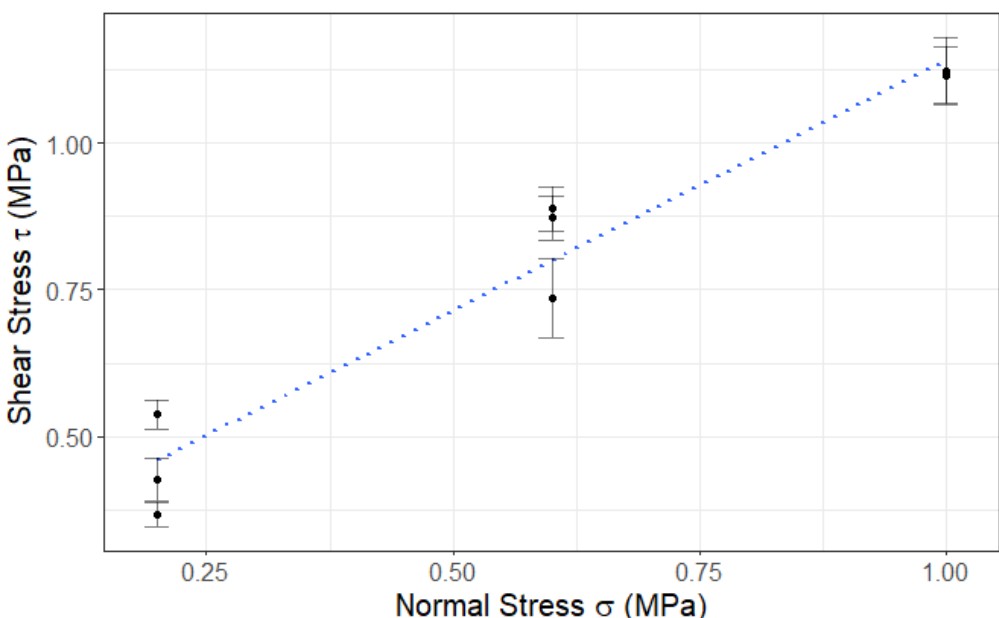

**Figure 8.** Normal stress versus shear stress for masonry triplets with different mortars, using common linear regression.

## 5. Conclusions

In the context of sustainability, while lime is often compared with cement and other binders through the lens of environmental impact, from a reliability/serviceability behavior standpoint, the impact of varying their ratio is often studied only at the mortar level. This study, however, primarily focused on their ratio's impact on important mechanical properties at the masonry level. The influence of varying the lime–cement ratio on properties such as the cohesion, angles of internal friction, flexural strength, and E-modulus of masonry was assessed in a single, consistent experimental campaign, with an emphasis on the repeatability of the experiments and the use of mixes that are commonly used in the industry. The experimental results were also compared with the predictions/recommendations of Eurocode 6.

The results show that the effect of lime in the mortar on the mechanical properties of the masonry was not prominently evident. Three mortar mixes were tested in the experiments: L67 (1:2:9), L50 (1:1:6), and Ref (1:0:5).

Increasing the quantity of lime in the binder led to a minor decrease in the compressive strength and E-modulus of the masonry. However, it resulted in a minor increase in the vertical deformation capacity of the masonry at peak load. The compressive strength of the masonry for all three mortars was found to range from 6.0 to 7.2 MPa, and the E-moduli ranged from 3.8 to 4.5 GPa. Furthermore, it was found that Eurocode 6 significantly overestimated (50–70%) the values of compressive strength. The flexural strength of the masonry was found to range from 0.1 to 1.2 MPa and was impacted by the strength of the mortar, rather than the presence of lime in the mortar. Among the mortar mixes tested, L67 had the lowest strength at the mortar level and led to the lowest values of masonry flexural strength in both parallel and perpendicular directions. L50 and Ref, both of which had similar mortar strengths, seemed to result in almost identical values as far as the flexural strength of the masonry was concerned. All of the obtained experimental values were higher than those in the guidelines of Eurocode 6, with one exception: L67 in the parallel direction had a characteristic value that was lower than 0.1 MPa. Finally, the strength of the mortar also had a direct impact on the coefficient of friction (0.58–0.87) of the masonry. It was not possible to discern a pattern that related mortar strength with cohesion (0.29–0.41 MPa). The data indicated that all measured cohesion values were higher than the values recommended by Eurocode 6.

The authors would like to acknowledge the need for testing more mixes prior to the generalization of the results. Furthermore, more units need to be tested in combination with the mixes tested herein to better understand the impact of the type of unit on the masonry, as well as the interaction of lime–cement mortars with different units such as stone and CMU blocks, specifically for shear bond strength. The authors hope that the results of the current research regarding the impact of lime on the mechanical performance of masonry will add to a more holistic understanding of sustainable constructions and the sustainable development of infrastructure and the preservation of the world's heritage, as outlined in the 2030 Agenda for Sustainable Development by the UN.

**Author Contributions:** Conceptualization, M.R., M.A. and P.B.L.; Methodology, M.R., M.A. and P.B.L.; Validation, M.R., M.A. and P.B.L.; Formal analysis, M.R., M.A. and P.B.L.; Investigation, M.R. and M.P.; Data curation, M.R.; Writing—original draft, M.R. and M.P.; Writing—review & editing, M.R., M.P., M.A. and P.B.L.; Visualization, M.R., M.P., M.A. and P.B.L.; Supervision, M.A. and P.B.L. All authors have read and agreed to the published version of the manuscript.

**Funding:** This work was partly financed by FCT/MCTES through national funds (PIDDAC) under the R&D Unit Institute for Sustainability and Innovation in Structural Engineering (ISISE), under reference UIDB/04029/2020, and under the Associate Laboratory Advanced Production and Intelligent Systems ARISE under reference LA/P/0112/2020. The authors gratefully acknowledge the European Lime Association for funding this project. The funding provided by the Portuguese Foundation for Science and Technology (FCT) to the research project PTDC/ECM-EST/1056/2014 (POCI-01-0145-FEDER-016841), as well to the research unit ISISE (POCI-01-0145-FEDER-007633) and the scholarship SFRH/BD/137358/2018, is also gratefully acknowledged.

**Institutional Review Board Statement:** Not applicable.

**Informed Consent Statement:** Not applicable.

**Data Availability Statement:** Not applicable.

**Acknowledgments:** The authors acknowledge the contributions of their colleague, Luciano Sambataro, for supplying supporting information that was relevant to the preparation of this paper.

**Conflicts of Interest:** The authors declare no conflict of interest.

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
