# Peer review of "Influence of Lime on Strength of Structural Unreinforced Masonry: Toward Improved Sustainability in Masonry Mortars"

_sustainability, doi:10.3390/su152115320_

Round 1

Reviewer 1 Report

All of the following comments should be addressed in detail and responded to in an appropriate manner so I can recommend the paper for publication:

1.     Keywords should be compressive strength or flexural strength

2.     If the author defines it as a sustainable mortar, then the author must perform an environmental assessment on it.

3.     Authors must establish a link between this manuscript and the journal.

4.     The manuscript should be very carefully checked to avoid any errors. The language should be checked throughout the text and any grammar mistakes should be corrected.

5.     The introduction does not sufficiently contextualize the study problem and the knowledge gap. The central objective of the research is also unclear. What is the current state of the art? What is the gap to be investigated? What is the contribution of the paper?

6.     The novelty of the article should be highlighted at the end of the introduction. There are several studies with a similar theme.

7.     Please add error bars to the figures. Please do this for the rest of the figures.

8.     Please make sure your conclusions' section underscore the scientific value added of your paper, and/or the applicability of your findings/results, as indicated previously. Please revise your conclusion part into more details. Basically, you should enhance your contributions, limitations, underscore the scientific value added of your paper, and/or the applicability of your findings/results and future study in this session.

All of the following comments should be addressed in detail and responded to in an appropriate manner so I can recommend the paper for publication:

1.     Keywords should be compressive strength or flexural strength

2.     If the author defines it as a sustainable mortar, then the author must perform an environmental assessment on it.

3.     Authors must establish a link between this manuscript and the journal.

4.     The manuscript should be very carefully checked to avoid any errors. The language should be checked throughout the text and any grammar mistakes should be corrected.

5.     The introduction does not sufficiently contextualize the study problem and the knowledge gap. The central objective of the research is also unclear. What is the current state of the art? What is the gap to be investigated? What is the contribution of the paper?

6.     The novelty of the article should be highlighted at the end of the introduction. There are several studies with a similar theme.

7.     Please add error bars to the figures. Please do this for the rest of the figures.

8.     Please make sure your conclusions' section underscore the scientific value added of your paper, and/or the applicability of your findings/results, as indicated previously. Please revise your conclusion part into more details. Basically, you should enhance your contributions, limitations, underscore the scientific value added of your paper, and/or the applicability of your findings/results and future study in this session.

Reviewer 2 Report

Selection of mortar for structural masonry is essential not only for achieving a durable structure with satisfactory service life, but also for improving its sustainability potential, as well as increasing the percentage of lime in the mortar will also result in a better performance environmentally, this paper details an experimental campaign to measure the compressive strength, E-modulus, flexural strength and shear bond strength of masonry samples containing two distinct lime-cement mortars and one cement mortar, aiming at quantitatively assessing the effect of a partial replacement of cement with lime in mortar at the masonry level. This study is of engineering reference value. However, several points should be addressed or discussed. Some remarks and questions are given below:

1.There seems to be a contradiction between the first conclusion in line 19 and the third conclusion in lines 21-23? Please check.

2.Since there is only qualitative description of test results in the Abstract, the authors should supplement the main quantitative results in the Abstract according to the Section 5 Conclusion.

In addition , the main conclusions for flexural strength test and shear bond strength test should supplement in the Abstract.

3.In section 2.3 Mortar mixes: preparation and characterization, three mortar mixtures were adopted for the test. The authors should add the reason why choose mix proportion of L50, L67 and Ref. If choose other mortar mixes proportion, Do the test conclusions change?

4.This paper only presents mechanics test result of compressive strength, flexural strength, and shear bond strength. The necessary mechanism analysis for the effect of lime on strength should be supplemented.

5.The authors should check the table carefully according to the format of the Journal. For example, check the upper and lower of character, bold or not?

In addition, in section 2.3, the text in lines 187 to 213 are bolded. The authors should check the paper carefully.

6.Some of the figures include several figures, eg. Fig.1~5,  Fig.7, etc. For the convenience of readers, the above-mentioned figures should add sub-title of figure.

7.Lines 335,340,351 etc. appear "Error! Reference source not found", the author should check the submitted paper carefully.

The quality of English Language is ok.

Reviewer 3 Report

Despite being extensively studied for its effects on mechanical properties of mortar, not much is known about the impact of binders on masonry as a whole. But, notwithstanding its environmental advantages, this study found that the effect of lime on the mechanical properties of masonry was not prominently evident when it was used as part of the binder. In most of the introduction, many references were presented to highlight the purpose of the study. However, the actual purpose of the study was not highlighted and is later mentioned in the conclusion. I think that the content succinctly is not described and contextualized with respect to previous and present theoretical background and empirical research on the topic.

This article details an experimental campaign to measure the compressive strength, E-modulus, flexural strength and shear bond strength of masonry samples containing two distinct lime-cement mortars (1:2:9 and 1:1:6, cement:lime:sand) and one cement mortar (1:0:5). The experimental campaign is aimed filling the knowledge gap on the effect of lime-incorporating mortars at the masonry level, including tested properties such as compressive strength, E- modulus under cyclic compression, flexural strength in different directions and shear bond strength. The tests are aimed at quantitatively assessing the effect of a partial replacement of cement with lime in mortar at the masonry level. The experiments showed that masonry's compressive strength when using three different mortars (L67, L50 and Ref) was in the 6.0 to 7.2 MPa range, and the E-modulus ranged from 3.8 to 4.5 GPa.  I think that the research design, questions, hypotheses and methods are not clearly stated and the arguments and discussion of findings coherent, are not balanced and compelling.

This work showed that the mortar's compressive strength was a reliable predictor of the masonry's flexural and shear bond strength. With regards to compression, it was found that higher levels of lime in mortar caused a marginal reduction in strength and stiffness, whilst slightly increasing pre-peak ductility of the masonry. The modulus of elasticity to compressive strength ratio for all the mortars was nearly identical, ranging from 600-650. most identical values as far as the flexural strength of masonry is concerned. It was not possible to discern a pattern that related mortar strength with cohesion. the results are not clearly presented and the conclusions thoroughly are not supported by the results presented in the article or referenced in secondary literature.

Round 2

Reviewer 3 Report

I think that the content succinctly is described and contextualized with respect to previous and present theoretical background and empirical research on the topic.

This research design, questions, hypotheses and methods are clearly stated and the arguments and discussion of findings coherent, are balanced and compelling.

The results are clearly presented and the conclusions thoroughly are  supported by the results presented in the article or referenced in secondary literature.